# Gerstmann–Sträussler–Scheinker Disease with F198S Mutation Induces Independent Tau and Prion Protein Pathologies in Bank Voles

**DOI:** 10.3390/biom12101537

**Published:** 2022-10-21

**Authors:** Rosalia Bruno, Laura Pirisinu, Geraldina Riccardi, Claudia D’Agostino, Elena De Cecco, Giuseppe Legname, Franco Cardone, Pierluigi Gambetti, Romolo Nonno, Umberto Agrimi, Michele Angelo Di Bari

**Affiliations:** 1Department of Food Safety, Nutrition and Veterinary Public Health, Istituto Superiore di Sanita’, 00161 Rome, Italy; 2Laboratory of Prion Biology, Department of Neuroscience, Scuola Internazionale Superiore di Studi Avanzati (SISSA), 34136 Trieste, Italy; 3Department of Neuroscience, Istituto Superiore di Sanita’, 00161 Rome, Italy; 4Department of Pathology, Case Western Reserve University, Cleveland, OH 44106, USA

**Keywords:** Gerstmann–Sträussler–Scheinker, tau, Alzheimer’s disease, bank vole, prion-like properties, prion

## Abstract

Gerstmann–Sträussler–Scheinker disease (GSS) is a rare genetic prion disease. A large GSS kindred linked to the serine-for-phenylalanine substitution at codon 198 of the prion protein gene (GSS-F198S) is characterized by conspicuous accumulation of prion protein (PrP)-amyloid deposits and neurofibrillary tangles. Recently, we demonstrated the transmissibility of GSS-F198S prions to bank vole carrying isoleucine at 109 PrP codon (BvI). Here we investigated: (i) the transmissibility of GSS-F198S prions to voles carrying methionine at codon 109 (BvM); (ii) the induction of hyperphosphorylated Tau (pTau) in two vole lines, and (iii) compared the phenotype of GSS-F198S-induced pTau with pTau induced in BvM following intracerebral inoculation of a familial Alzheimer’s disease case carrying Presenilin 1 mutation (fAD-PS1). We did not detect prion transmission to BvM, despite the high susceptibility of BvI previously observed. Immunohistochemistry established the presence of induced pTau depositions in vole brains that were not affected by prions. Furthermore, the phenotype of pTau deposits in vole brains was similar in GSS-F198S and fAD-PS1. Overall, results suggest that, regardless of the cause of pTau deposition and its relationship with PrP^Sc^ in GSS-F198S human-affected brains, the two components possess their own seeding properties, and that pTau deposition is similarly induced by GSS-F198S and fAD-PS1.

## 1. Introductions

Prion diseases or transmissible spongiform encephalopathies represent a group of neurodegenerative disorders that affect humans and animals. They are associated with the accumulation of a misfolded form of the cellular prion protein (PrP^C^) called scrapie prion protein (PrP^Sc^). Prion diseases are remarkably transmissible to susceptible hosts in natural and experimental conditions.

Gerstmann–Sträussler–Scheinker disease (GSS) is a rare, dominantly-inherited neurodegenerative disorder associated with mutations in the prion protein gene (*PRNP*). The first mutation reported was a substitution of leucine for proline at codon 102 (P102L) of cellular prion protein (PrP^C^); since then, at least 26 additional mutations have been described [1]. Among these, the missense mutation leading to an amino acid substitution of phenylalanine with serine at codon 198 of the PrP^C^ (F198S) was first reported in a large Indiana kindred in 1992 [2].

GSS-F198S-affected brains are characterized by PrP^Sc^ plaque depositions, involving cerebrum and cerebellum [3]. A striking feature of this disease is the presence of hyperphosphorylated Tau (pTau) and numerous neurofibrillary tangles (NFTs) made of intraneuronal filamentous Tau aggregates, both present in the grey matter of the cerebrum [3,4]. PrP^Sc^ association with pTau in GSS-F198S brain patients does not appear to be casual, but its histogenesis is poorly understood. The presence of pTau was also associated with other GSS genotypes, such as P105L, A117V, Q217R [5], and more rarely in P102L [6]. Some authors have suggested that Tau-related pathology could be a possible end stretch of prion-induced neurodegeneration [7], yet PrP^Sc^ deposits precede the development of Tau pathology [8].

Morphologically, NFTs of GSS-F198S are identical to those observed in Alzheimer’s disease (AD) [5,9]. Recently, this observation has been confirmed by a study of atomic models of paired helical filaments (PHFs), in which Tau filaments obtained from GSS-F198S resulted identical to those from AD [10]. Interestingly, the co-presence of pTau aggregates with PrP^Sc^ in the brain of GSS-F198S-affected patients is quite reminiscent of what occurred in AD-affected brains, in which Tau filaments accumulate with Aß-amyloid. The presence of Tau during these two distinct amyloidoses suggests the activation of a common amyloid-dependent mechanism resulting in pTau formation [10]. Interestingly, in recent years, AD, tauopathies and several other proteinopathies were compared to prion diseases for their ability to self-propagate, thereby extending the prion concept [11,12]. Particularly for AD and several tauopathies, the prion-like experimental induction and propagation of Tau pathology were widely demonstrated using either transgenic or wild-type mice [13,14].

In the past two decades, we leveraged the bank vole (*Myodes glareolus*) as a susceptible wild-type animal model for transmission studies of human and animal prions focusing in two lines, BvI and BvM, that respectively carried isoleucine and methionine at codon 109 of PrP and showed specific sensitivity to different prions [15,16,17,18,19,20,21,22,23,24,25].

In a previous study [20], we demonstrated the efficient transmission of human GSS-F198S prions to BvI after intracerebral inoculation, which was associated with a specific pathological phenotype. The present work extends our study to the BvM line, to compare the susceptibility of both vole lines to GSS-F198S prions. Remarkably, we did not detect prion transmission to BvM, despite the high susceptibility of BvI previously observed [20].

However, the previous work focused only on the effect of prions in recipient BvI irrespective of the co-presence of pTau in the inoculum. Here, we extended our investigation to the effect of pTau presence in GSS-F198S homogenates on the recipient brain of both genetic vole lines. Strikingly, neo-formed pTau depositions were observed in the brain of voles that survived to prions, irrespective of the vole genotype. Remarkably, no mixed deposition of pTau and PrP^Sc^ in the same vole brain was detected.

Finally, we compared the phenotype of GSS-F198S-induced pTau with pTau deposits induced in BvM following intracerebral inoculation of a familial AD case carrying Presenilin 1 mutation (fAD-PS1). We found that the shape and distribution of pTau deposits in vole brains were similar in both GSS-F198S and fAD-PS1.

## 2. Materials and Methods

### 2.1. Ethics Statement

Voles were obtained from the breeding colony at the Istituto Superiore di Sanita’ (ISS). The research protocol, approved with authorization number 1119/2015-PR by the Service for Biotechnology and Animal Welfare of the ISS and authorized by the Italian Ministry of Health, adhered to the guidelines included in the Italian Legislative Decree 116/92, which transposed the European Directive 86/609/EEC on Laboratory Animal Protection, and in the Legislative Decree 26/2014, which transposed the European Directive 2010/63/UE on Laboratory Animal Protection. All animals were individually identified by a passive integrated transponder (Bayer) [16,18].

The use of human autopsy material from cases of GSS-F198S was approved, as previously reported in Pirisinu et al. 2016. fAD-PS1 and non-demented patients were referred to the Italian National Registry of Creutzfeldt–Jakob disease (CJD) and related syndromes at the ISS with a clinical suspect of CJD.

### 2.2. Inocula

Human brain homogenates from subjects affected with GSS-F198S and familial AD were used for this study. The two GSS-F198S patients were brothers carrying the same GSS mutation but a distinct haplotype at codon 129: VV and MV, respectively, and herein and in Pirisinu et al. 2016, referred to as case #3 and #4. Age at onset of disease, duration and clinical signs are reported in Pirisinu et al. [20]. Both patients showed similar neuropathological features, characterized by multicore plaques of PrP^Sc^ and neurofibrillary tangles (NFT). Fine spongiform degeneration was observed only in case #4. Homogenates used in this work were the same as those previously used in Pirisinu et al. [20] and were obtained from frontal cortex prepared as 10% *w*/*v* in phosphate-buffered saline (PBS) and stored at −80 °C.

The frontal cortex of a neuropathologically confirmed case of familial AD with Presenilin 1 with S169L mutation (fAD-PS1) and temporo–occipital cortex of an elderly healthy patient were used to prepare homogenates at 20% *w*/*v* in PBS

### 2.3. Bioassays, Clinical Examination and Sampling

Groups of eight-week-old BvM were intracerebrally inoculated free-hand under ketamine anesthesia (ketamine 0.1 mg/g), with 20 µL of human brain homogenates into the left cerebral hemisphere frontally, in the frontal motor cortex and hippocampus. All voles were clinically examined twice a week for the appearance of neurological signs like a motor or cognitive deficit until the end of the experiment. Animals were culled in a 100% saturated carbon dioxide room. Immediately at post-mortem, each brain was removed and divided into two parts by a sagittal paramedian cut [18]. The smaller portion (left part) was immediately stored at −80 °C for biochemical investigation, while the larger one (right part) was fixed in 10% neutral buffered formalin [26].

### 2.4. Histology, Immunohistochemistry and Biochemistry

Fixed brains were trimmed at standard coronal levels, embedded in paraffin wax and cut in sections of 5 µm. Sections were stained with hematoxylin and eosin staining for histopathological analysis. For pTau immunohistochemistry sections upon hydration were autoclaved in 0.2% of citrate buffer solution at pH 6.2 for 15 min at 98 °C. Endogenous peroxidase activity was inhibited by immersion of the sections in 3% hydrogen peroxide for 20 min. After blocking with 6% normal goat serum for 30 min, sections were incubated with primary antibody overnight at 4 °C. Following secondary antibody incubation, immunoreactivity was detected by the avidin–biotin-complex (Vector) method as suggested by suppliers. Sections were stained with diaminobenzidine and counterstained with Mayer’s hematoxylin. Each IHC run included positive and negative control sections.

A panel of anti-Tau antibodies was used. AT8 antibody recognizes hyperphosphorylated epitopes at residues Ser202 and Thr205 (Thermo-Scientific, Waltham, MA, USA). AT180 and PHF-1 are phosphorylation-dependent anti-Tau Abs that recognize epitopes at residues Thr231 and Ser396-Ser404, respectively (AnaSpec, Fremont, CA, USA). Non-phosphorylation-dependent anti-human tau monoclonal antibody for amino acids 141–178, T14 (Novex, Mumbai, India), was used to discriminate human and vole Tau in the brains of inoculated voles.

Immunohistochemistry and western-blot analysis for PrP^Sc^ were performed as previously described [20].

### 2.5. Sequence of Tau Gene

Total RNA was isolated by manually homogenizing bank vole brains with micro pestles (Kisker Biotech GmbH, Steinfurt, Germany) in TRIzol (Invitrogen, Waltham, MA, USA). RNA isolation was performed using PureLinkTM RNA Mini kit (Thermo Fisher Scientific, Waltham, MA, USA) following the manufacturer’s instructions. mRNA fraction was retrotranscribed using SuperScript^TM^ Reverse Transcriptase (Thermo Fisher Scientific). Primers for amplification of the tau gene were manually designed by aligning the corresponding murine cDNA sequences (reference genes: Gene ID 17762, for tau) with bank vole genomic contigs.

The following primers were used for Tau amplification: forward (5′–3′) GAACCAGTATGGCTGAACCCC; reverse (5′–3′) TGATCACAAACCCTGCTTAGCC. To amplify Tau sequence, PCR reactions were prepared using Phusion^®^ High. Fidelity DNA Polymerase (NEB) followed the supplier’s instructions. Amplicons corresponding to tau (around 1500 bp, as calculated from the corresponding murine sequences) were extracted and sent for sequencing.

## 3. Results and Discussion

### 3.1. Resistance of BvM to GSS-F198S Prions

In a previous study [20], we observed high susceptibility of BvI to GSS-F198S prions. In particular, we inoculated two brain homogenates obtained from two patients, named #3 and #4, observing high and low attack rates, respectively, as reported in Pirisinu et al. [20]. Transmission data were in accordance with the amount of PrP^Sc^ in the inoculum [20].

In the present study, we inoculated the brain homogenates of case #3 and #4, in two groups of 13 and 12 BvM, respectively. BvM didn’t show obvious clinical signs and were sacrificed for intercurrent disease or died at typical ages for uninoculated voles until 1047 days post-inoculation (d.p.i.). In particular, BvM showed survival times of 225–939 and 103–1047 d.p.i., respectively, for case #3 and #4. Ten voles from each group were culled, and both formalin-fixed and frozen half brains were collected. The remaining voles of each group were found dead, and the brains were collected only for freezing. All fixed brains, analyzed using Hematoxylin and Eosin staining, did not show neuropathological changes such as spongiosis, neuronal loss, or gliosis. Immunohistochemistry for PrP^Sc^ was performed on all fixed brains, and none of the brains showed immunolabelling. Furthermore, all frozen brains were analyzed for research of PrP^Sc^ by western-blot, and all were found negative for PrP^Sc^ (Appendix A).

Interestingly, although we used the same inocula, we found that GSS-F198S prions did not replicate in BvM, in contrast with the high susceptibility observed in BvI.

### 3.2. Propagation of GSS-F198S-Derived pTau in Bank Voles Is Independent of GSS-F198S Prions

In order to investigate whether intracerebral inoculation of GSS-F198S brain homogenates containing both pTau and PrP^Sc^ was also competent to induce pTau in recipient voles, we examined vole brains for the presence of induced pTau depositions. In particular, we examined the presence of induced pTau depositions in BvI and BvM inoculated with both GSS-F198S cases, either PrP^Sc^-positive or PrP^Sc^-negative, for which the formalin-fixed brains were available. Furthermore, we also included as a negative control, a group of 21 BvM inoculated with an elderly healthy human brain homogenate (Hu-Ctrl) and a group of 30 uninoculated voles consisting of 18 BvM and 12 BvI. More specifically, voles inoculated with Hu-Ctrl were culled at 365 (*n* = 4), 540 (*n* = 7) and 810 (*n* = 10) d.p.i., while uninoculated voles were culled at 725–1204 days. pTau-induced depositions were searched by immunohistochemistry, using a panel of common anti-Tau antibodies, recognizing pathological hyperphosphorylated epitopes. These antibodies were selected based on the vole Tau protein sequence shown here for the first time (Figure 1).

Fourteen out of fifteen BvI inoculated with case #3 were sick with GSS-F198S prions in short times of 85–183 d.p.i. [20]; of fourteen PrP^Sc^-positive brains, six were analyzed for pTau deposition and were found negative. The single survivor to prions was culled at 524 d.p.i. and resulted negative for pTau. Otherwise, for case #4, only one out of thirteen BvI was sick for GSS-F198S prions at 153 d.p.i. and was pTau-negative. Of twelve survivors and PrP^Sc^-negative, four were analyzed for pTau showing depositions in the brain. Remarkably, no mixed deposition of pTau and PrP^Sc^ in the same vole brain was detected.

Conversely, from BvM transmission we observed that of thirteen voles inoculated with case #3 and resulting in PrP^Sc^-negative, ten were analyzed for pTau of which nine were found positive (225–939) and one was pTau-negative (669 d.p.i.). Similarly, of twelve BvM inoculated with GSS-F198S case #4 and resulting in PrP^Sc^-negative, ten were analyzed for pTau depositions, of which nine were pTau-positive (417–1047) and one did not show pTau deposition (103 d.p.i.). 

Deposition of induced vole pTau (bv-pTau) was mainly confined in the alveus and showed two specific patterns: neuropil threads and oligodendroglial coiled bodies (Figure 2). Rarely, bv-pTau deposits were found intraneuronal in the hippocampus, entorhinal cortex and amygdala (Figure 3). Both vole lines showed the same pTau deposition patterns. The deposition patterns observed in voles were similar to those observed in laboratory wild-type mice after inoculation with sporadic tauopathies, previously described by Clavaguera and colleagues [14]. The presence of human pTau residues of inoculum in the pTau-positive vole brains was excluded using a human T14 tau-specific antibody (Appendix A).

These findings demonstrate for the first time that human pTau from GSS-F198S propagates in vole brains after intracerebral inoculation. More importantly, the present study shows that propagation of bv-pTau is independent from the PrP^Sc^ infection, suggesting that bv-pTau deposition is directly triggered by human pTau. In fact, we found that pTau was present in the PrP^Sc^-negative voles surviving longer than 225 d.p.i. indicating that both human proteinopathies can propagate independently and that human pTau can preserve its own seeding properties independently of prions. In spite of the proximity of Tau depositions and PrP^Sc^ deposits in the natural host [3,4,8], the different propagation in the recipient animal model highlights that the specific seeding activity of each protein is conserved. These data did not exclude that in the human brain pTau is induced by PrP^Sc^ formation in the human brain but suggested a specific seeding activity of pTau independent of its origin. Thus, the selective transmission of PrP^Sc^ and pTau proteinopathies from GSS-F198S can be achieved in voles. However, previous studies led to the conclusion that pTau deposition is a consequence of prion infection and is found in proximity of PrP^Sc^ deposits [27,28].

Interestingly, we did not find the compresence of both pTau and PrP^Sc^ deposits in the same vole brain, probably due to different propagation times. In fact, prions spread in voles more rapidly than pTau. These data are reminiscent of what has been observed in adult individuals that died for CJD contracted during their childhood by treatment performed with cadaveric human growth hormone containing prions (iatrogenic CJD). Some relatively young age cases showed the presence of amyloid-β amorphous aggregates probably caused by Aß seeds transmitted during the iatrogenic procedures independently from the PrP^Sc^ seeding process [29,30,31,32]. The silent presence of Aβ pathology in iCJD cases can be due to the rapid replication of prions in comparison to Aβ or by different seeding properties.

### 3.3. FAD-PS1 Induce pTau Pathology in Voles Similarly to GSS-F198S

In order to investigate the pTau induction properties of other human tauopathies, we then investigated a group of BvM inoculated with a familial AD carrying Presenilin 1 mutation (fAD-PS1) case. In particular, we intracerebrally inoculated a group of fifteen BvM with fAD-PS1 brain homogenate. Thirty out of fifteen voles were serially culled at 180 (*n* = 2), 365 (*n* = 4), and 810 (*n* = 7) d.p.i., while two voles were sacrificed respectively at 540 and 720 d.p.i for intercurrent disease. Inoculated voles remained free of neurological signs. Deposition of bv-pTau was observed in thirteen out of fifteen voles (the two pTau-negatives voles were among those culled at 810 d.p.i.).

The bv-pTau species from both human sources, i.e., GSS-F198S and fAD-PS1, were compared as to immunoreactivity, topographic and morphologic characteristics of the induced deposits and no differences as to these parameters were observed (Figure 4 and Appendix A). Interestingly, we observed that voles culled at 180 d.p.i. displayed a low amount of bv-pTau deposits while voles culled or that succumbed successively showed high deposition density (Appendix A). The same observation was found in the brain of voles inoculated with GSS-F198S and culled at 225 d.p.i. in comparison to the voles culled latterly. Furthermore, at the end of GSS-F198S and fAD experimental transmissions, vole brains showed a variable quantity of Bv-pTau deposits in the alveus. Although the experiments were not harvested at precise time points after inoculation, our data suggest a progressive, but not linear accumulation related to time. Thus, the high variability of both the density and the number of pTau deposits caused difficulties to perform a quantitative assessment.

Numerous neuropathological studies [5,9,33] have demonstrated that the substructure of the neurofibrillary lesions of GSS-F198S are indistinguishable from those present in AD. Interestingly, recently, Hallinan and coworkers by cryo-EM revealed that the atomic models have shown that Tau filaments of patients affected with GSS-F198S are identical to those isolated from AD [10].

Here, we found that voles are competent for inducing resident neo-formed pTau after exposition to human Tau of fAD-PS1 and GSS-F198S. As already observed for GSS-F198S, induced-pTau from fAD-PS1 did not cause obvious neurological symptoms during the routine clinical observation. Furthermore, the comparison of bv-pTau deposits induced by both human sources highlights that neo-formed Tau depositions had the same immunoreactivity and morphological features.

## 4. Conclusions

In this pilot study, we show for the first time that human pTau from the genetic prion disease GSS-F198S and the genetic fAD-PS1 can propagate in bank voles. Furthermore, independently of the cause of pTau deposition and its relationship with PrP^Sc^ in GSS-F198S human-affected brains, we observed that the two components possess their own seeding properties. In general, to our knowledge, this is the first time that the two components coexisting in a single source have been independently transmitted to a wild-type host, offering a promising model for gaining insights into the pathogenic mechanisms of human neurodegenerative proteinopathies. Overall, our findings strongly suggest that coexisting misfolded proteins, i.e., PrP^Sc^ and pTau, in the same brain of patients affected with GSS-F198S, although pathogenetically correlated, can conserve their own seeding properties. More importantly, although our observations were supported using two GSS-F198S cases and the results are consistent, further transmission studies of other GSS cases accumulating both PrP^Sc^ and pTau will be useful for extending and further confirming such evidence.

Further biochemical study will be performed on Tau and pTau in solution using WB to study the biochemical properties of induced bv-pTau aggregates. To deepen the prion-like propagation properties of neo-formed bv-pTau in the same host, we will be undertaking inoculation of pTau-positive vole brains into naive voles.

In the present study, we further investigated the in vivo prion-like propagation properties of human pTau assemblies from GSS-F198S and fAD-PS1, and we observed that pTau induced in bank vole brains had the same immunoreactivity and morphological features. Given the encouraging susceptibility obtained with PrP^Sc^ and pTau using the bank vole model, the presence of potential neo-formed Aβ depositions in recipient vole brains following inoculation with fAD-PS1 will be investigated.

Overall, these data are helpful for further investigating the molecular mechanisms that underlie the pathogenesis of some amyloidogenic disorders, such as prion diseases or Alzheimer’s disease, and the involvement of Tau protein as a co-actor of the pathogenic processes.

## Figures and Tables

**Figure 1 biomolecules-12-01537-f001:**
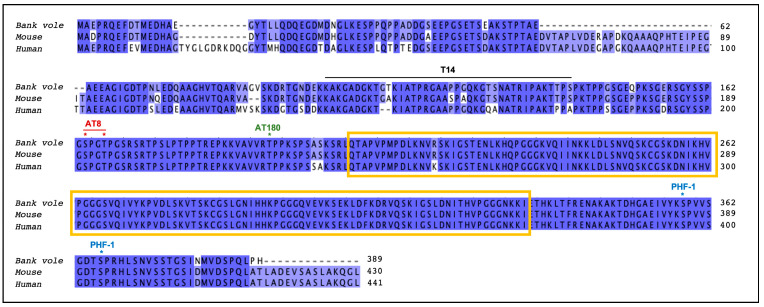
Alignment of bank vole, murine, and human Tau proteins. Colored boxes highlight highly conserved (dark blue), poorly conserved (light blue) and not conserved (white) residues. Yellow boxes indicate the microtubule binding region of Tau. For the alignment, the longest mouse Tau (430 amino acids) and human Tau (441 amino acids) isoforms were chosen. Epitopes recognized by the antibodies used in this study have been mapped on the aligned sequences using human Tau as reference.

**Figure 2 biomolecules-12-01537-f002:**
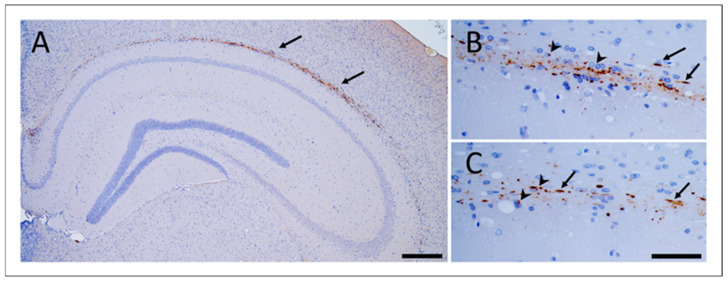
Tau pathology in bank voles inoculated with GSS-F198S. Hyperphosphorylated Tau aggregates in the brain of a BvM culled at 442 d.p.i., inoculated with GSS-F198S #3. Tau inclusions accumulated along the alveus (picture (**A**), scale bar 200 µm). In pictures (**B**,**C**), hyperphosphorylated Tau depositions are shown as neuropil threads (arrows) and coiled bodies (head arrows) (scale bar 50 µm). Immunostaining was performed with AT8 antibody.

**Figure 3 biomolecules-12-01537-f003:**
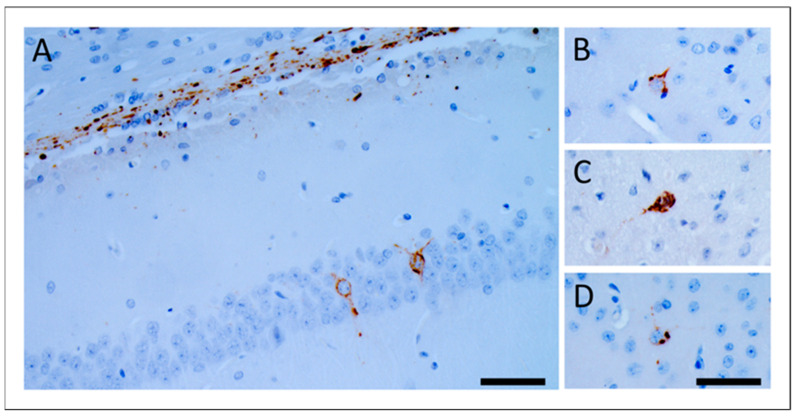
Intraneuronal bv-pTau deposits in voles inoculated with GSS-F198S. Immunoreactive neuronal bodies observed in hippocampus (**A**), entorhinal cortex (**B**), amygdala (**C**), and parietal cortex (**D**). Immuno-staining was performed with AT8 antibody. Scale bars 20 µm.

**Figure 4 biomolecules-12-01537-f004:**
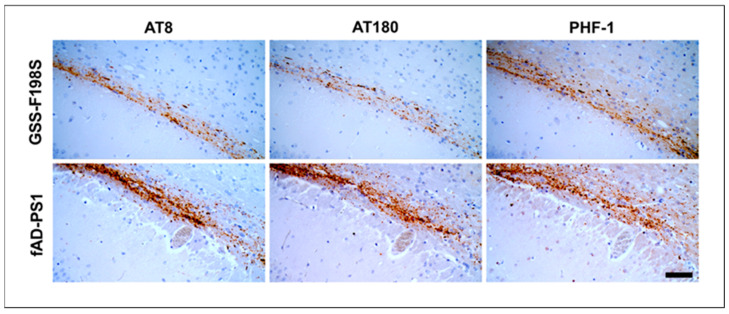
Filamentous Tau pathology in the alveus of BvM inoculated with GSS-F198S and fAD-PS1. Using different anti-Tau antibodies, AT8, AT180, and PHF-1, respectively, against Ser202-Thr205, pT231 and pS396/pS404 epitopes, the same pTau deposition pattern was observed. Scale bar 20 µm.

## Data Availability

Data is contained within the article or Appendix A.

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
