# Peer review of "Gerstmann–Sträussler–Scheinker Disease with F198S Mutation Induces Independent Tau and Prion Protein Pathologies in Bank Voles"

_biomolecules, 2022, doi:10.3390/biom12101537_

Round 1

Reviewer 1 Report

Bruno et al "Gerstmann-Straussler-Scheinker disease with F198S mutation: induced hyperphosphorylated-tau in bank vole is independent of prions and morphologically analogous to Alzheimer's disease"

In this work, the authors present evidence that injecting the human fAD brain crude homogenates to the bank vole brain can produce Abeta and pTau pathology but not neurological abnormalities in the animals. The deposition of these human amyloids is apparently independent of the bank vole prion subtypes, contrarily to prion propagation. The authors suggested that bank vole affords a good “wildtype” animal model to studying the seeding and propagation of AD tau fibrils.

In general, this is an interesting work, much of which was built on a previous report by the same group, using a different PrP subtype. While intriguing and potentially useful, the insight obtained from this work appears to be incremental. The authors are commended for their long strive to develop a bank vole model for human CNS amyloidosis. However, the innovation element of the current work is limited, except the use of a different rodent model. It is unclear as to whether the majority of AD, sporadic and late-onset, can be recapitulated in this model. Even so, the conspicuous Abeta and tau pathologies without discernible neurological manifests further erode the argument for a new and faithful disease model. In addition, the use of crude extracts from the brain is problematic, which is far from be sufficient to provide a clear-cut insight on the contribution of tau, Abeta, PrP, or any other brain factors (e.g., inflammatory molecules) associated with the underlying pathology. The enthusiasm for this manuscript is modest.

Author Response

In this work, the authors present evidence that injecting the human fAD brain crude homogenates to the bank vole brain can produce Abeta and pTau pathology but not neurological abnormalities in the animals. The deposition of these human amyloids is apparently independent of the bank vole prion subtypes, contrarily to prion propagation. The authors suggested that bank vole affords a good “wildtype” animal model to studying the seeding and propagation of AD tau fibrils.

In general, this is an interesting work, much of which was built on a previous report by the same group, using a different PrP subtype. While intriguing and potentially useful, the insight obtained from this work appears to be incremental. The authors are commended for their long strive to develop a bank vole model for human CNS amyloidosis. However, the innovation element of the current work is limited, except the use of a different rodent model. It is unclear as to whether the majority of AD, sporadic and late-onset, can be recapitulated in this model.

Even so, the conspicuous Abeta and tau pathologies without discernible neurological manifests further erode the argument for a new and faithful disease model. In addition, the use of crude extracts from the brain is problematic, which is far from be sufficient to provide a clear-cut insight on the contribution of tau, Abeta, PrP, or any other brain factors (e.g., inflammatory molecules) associated with the underlying pathology. The enthusiasm for this manuscript is modest.

Response: We thank the reviewer very much for pointing this out. A part of the limitations evidenced in our study is now widely discussed.

In this work, the use of the bank vole model is to prove a principle that Tau and PrPSc had distinct and own propagation properties. Despite being an imperfect model, it allowed us to achieve the replication data we observed. Further studies on Tau induction and replication are ongoing. Unfortunately, the lack of pathology, clinical signs and the slow pTau propagation, make these studies very complex and require a long time. We set the transmissions according to the commonly used protocol the for prion field and it is the elective method for bank vole and mice inoculation in our lab which was recently also used by Clavaguera et al. 2013. The crude extract mimics the conditions closest to an iatrogenic administration, although intracerebral inoculation still represents a forcing of the biological system.

Regarding sporadic AD, we have ongoing experimental transmissions using sporadic AD cases and other mutated cases for PS1 and PS2.

Reviewer 2 Report

The authors of this manuscript showed that a link between prion diseases or Alzheimer's disease may exist, and that Tau protein may play a role in the pathogenic processes. Thus, the human pTau from the genetic prion disease GSS-F198S and the genetic fAD-PS1 can propagate in voles. Misfolded proteins PrPSc and pTau may coexist in the same brain. These data are important to understand the molecular mechanisms of neurodegenerative diseases such as prion diseases and Alzheimer's disease.  Hyperphosphorylated Tau deposition was found  similarly induced by both GSS-F198S and fAD-PS1. Voles are able to replicate human Tau from fAD-PS1 and 276 GSS-F198S. The article is well written and of interest o the Journal’s readers. It can be published in the present form.

Author Response

The authors of this manuscript showed that a link between prion diseases or Alzheimer's disease may exist, and that Tau protein may play a role in the pathogenic processes. Thus, the human pTau from the genetic prion disease GSS-F198S and the genetic fAD-PS1 can propagate in voles. Misfolded proteins PrPSc and pTau may coexist in the same brain. These data are important to understand the molecular mechanisms of neurodegenerative diseases such as prion diseases and Alzheimer's disease.  Hyperphosphorylated Tau deposition was found  similarly induced by both GSS-F198S and fAD-PS1. Voles are able to replicate human Tau from fAD-PS1 and 276 GSS-F198S. The article is well written and of interest o the Journal’s readers. It can be published in the present form.

Response: We thank the reviewer very much for his/her comment and for valuing our work. We modified the text according to the suggestions of the other reviewers, hoping to have further improved the information in the paper for the readers.

Reviewer 3 Report

The manuscript by Bruno et al describes a GSS-F198S prion protein resistant bank vole model (BvM), that is still somewhat susceptible to pTau pathology seeded from the patient brain homogenate. They compare this pathology with their previously described GSS model (BvI), which also shows prion pathology. Based on the vole ages and presence of Tau pathology in the BvM but not the BvI model, the authors conclude that Tau pathology is independently seeded in this animal model, and it is slower in progression than prion aggregation. Finally, they do similar analysis with fAD tissue inoculated voles, and find robust Tau pathology. The paper establishes that bank voles are suitable animal models for studying Tau seeding based pathology in addition to their prion disease modeling abilities.

In the current form of the manuscript, the claims appear overstated. The lack of quantitative analysis brings the question of reproducibility of pathology across animals.  The paper will require several additional experiments and analysis to address the lacunae that I list below:

1.       Section 3.2 -Time-course: The authors have designed their experiments conducive for a preliminary time-course study progressive pathology. But without actual quantitative analysis of the extent of spread and/or intensity of pathology across time, it is unclear how long the Tau pathology takes to appear and if it is indeed progressive. I understand that brains were not harvested at precise time-points after inoculation, but they can be coarsely binned. This kind of analysis will add immense value to the paper and the understanding of the progression.

2.       The authors have collected frozen halves for biochemical processing. They should show that that the pTau and Tau antibodies that they have used work selectively in the bank vole brain tissue. This analysis would also potentially allow them to easily quantify extent of brain pathology across experimental samples.

3.       Fig 4: The difference in ta pathologies between the GSS-F198S and fAD-PS1 inoculated vole brains is stark. While the hyperphosphorylated Tau (AT8) and phospho-Tau S202/T205 (AT180) appear to be much higher, the PHF-1 staining is comparable. Again, I am basing this on the one set of representative images shown by the author. Does this suggest that the pathology of Tau phosphorylation is different between the conditions?

a.       Have the authors quantified the amount of total protein between the inocula (ln 111-113? Further, it is important to show the profile of total Tau and/or phosphorylation status both qualitatively and qualitatively using western blots or other measures.

b.       Is it possible that the amyloid component in the fAD inoculum leads to this accelerated pathology or only that Tau from the homogenate is responsible for the seeding?

4.       To make the claim that the Tau phosphorylation pathology seen in the vole brains initiated by patient inoculum seeding is truly mimicking that what is observed in GSS and/or AD brains, the authors should do a vole tissue inoculum study to see whether progression gets passed on.

5.       Ln 116-117: Which part of the cortex was used to prepare homogenates? Were protein concentrations measured? While the authors describe that the homogenates were injected into the left cerebral hemispheres, additional details about location need to be included. 20 ul is a large volume to be injected in a non-ventricular location. How did the authors prevent bolus related tissue damage?

Author Response

The manuscript by Bruno et al describes a GSS-F198S prion protein resistant bank vole model (BvM), that is still somewhat susceptible to pTau pathology seeded from the patient brain homogenate. They compare this pathology with their previously described GSS model (BvI), which also shows prion pathology. Based on the vole ages and presence of Tau pathology in the BvM but not the BvI model, the authors conclude that Tau pathology is independently seeded in this animal model, and it is slower in progression than prion aggregation. Finally, they do similar analysis with fAD tissue inoculated voles, and find robust Tau pathology. The paper establishes that bank voles are suitable animal models for studying Tau seeding based pathology in addition to their prion disease modeling abilities.

In the current form of the manuscript, the claims appear overstated. The lack of quantitative analysis brings the question of reproducibility of pathology across animals.  The paper will require several additional experiments and analysis to address the lacunae that I list below:

Point 1: Section 3.2 -Time-course: The authors have designed their experiments conducive for a preliminary time-course study progressive pathology. But without actual quantitative analysis of the extent of spread and/or intensity of pathology across time, it is unclear how long the Tau pathology takes to appear and if it is indeed progressive. I understand that brains were not harvested at precise time-points after inoculation, but they can be coarsely binned. This kind of analysis will add immense value to the paper and the understanding of the progression.

Response 1: We thank the reviewer for pointing this out. Yes, although the quantitative analysis of the extent of Bv-pTau spread and/or intensity of pathology across time is not an aim of the study, we observed less numerous deposits in the BvM culled early in comparison to voles culled after at the end of experimental transmission (as now add in lines 279-288). The vole that was inoculated with GSS and culled at 225 d.p.i. and in those voles inoculated with fAD and culled at 180 d.p.i. was displayed a low amount of Bv-pTau deposits while voles culled or succumbed successively showed high deposition density. Interestingly, at the end of all three experimental transmissions, vole brains showed a variable quantity of Bv-pTau deposits in the alveus. Although the experiments were not harvested at precise time points after inoculation, our data suggest a progressive, but not linear accumulation related to time.

We added a new figure (Figure S4) and modified the text accordingly, to better describe the bv-pTau amount in early and latter culled bank voles.

Point 2: The authors have collected frozen halves for biochemical processing. They should show that that the pTau and Tau antibodies that they have used work selectively in the bank vole brain tissue. This analysis would also potentially allow them to easily quantify extent of brain pathology across experimental samples.

Response 2: We thank the reviewer for pointing this out. We have not yet approached the study with biochemical methods. Unfortunately, we used the brain tissue collected from voles inoculated with GSS-F198S cases to perform the deepened diagnosis, and we have only some residual homogenates. In the discussion, we have added a sentence highlighting and specifying that further study will be performed to study the pTau and Tau in solution using WB (lines 327-328).

 Moreover, we determined the Abs for the detection of pTau based on the vole Tau protein sequence. We also confirmed the specificity of Abs against bv-pTau, using the human T14 tau-specific antibody that excluded the observation of human pTau residues of inoculum in the pTau-positive vole brains. Furthermore, the numerous uninoculated vole brains used as negative controls support this evidence.

Point 3:  Fig 4: The difference in ta pathologies between the GSS-F198S and fAD-PS1 inoculated vole brains is stark. While the hyperphosphorylated Tau (AT8) and phospho-Tau S202/T205 (AT180) appear to be much higher, the PHF-1 staining is comparable. Again, I am basing this on the one set of representative images shown by the author. Does this suggest that the pathology of Tau phosphorylation is different between the conditions?

  1. Have the authors quantified the amount of total protein between the inocula (ln 111-113? Further, it is important to show the profile of total Tau and/or phosphorylation status both qualitatively and qualitatively using western blots or other measures.
  2. Is it possible that the amyloid component in the fAD inoculum leads to this accelerated pathology or only that Tau from the homogenate is responsible for the seeding?

Response 3: Regarding the apparent variation observed by the reviewer in Figure 4, using different Abs, we observed that there is no significant variability in the pathology of Tau phosphorylation. Furthermore, in the image the PHF1 antibody caused background and the evaluation of the relative amount is less easy. Contrariwise, we observed high variability of both density and the numbers of pTau deposits,  causing difficulties in the quantitative evaluation. For this reason, we kept the assessment of the recognized patterns very rigorously.

As reported above, we observed variability of bv-pTau deposits in the brain at the end of the experiments. In addition, it is too difficult to compare the amount of pTau in the brains of vole inoculated with GSS and fAD, which were generated starting from different homogenate concentrations, 10% and 20%, respectively. The pictures were collected by taking cases from both groups and not selecting them. The figure is intended to compare only the accumulation patterns and not the quantity.

 Point 4: To make the claim that the Tau phosphorylation pathology seen in the vole brains initiated by patient inoculum seeding is truly mimicking that what is observed in GSS and/or AD brains, the authors should do a vole tissue inoculum study to see whether progression gets passed on.

Response 4: We thank the reviewer for pointing this out. To better address this point , we have  ongoing further experimental transmissions (lines 328-330). Our data suggests that the vole brains initiated by patient inoculum seeding seem to be mimicking what is observed in GSS and/or AD brains, and for this reason, we evidenced in the revision that we performed a pilot study (line 306, and discussion revisited)

Point 5:  Ln 116-117: Which part of the cortex was used to prepare homogenates? Were protein concentrations measured? While the authors describe that the homogenates were injected into the left cerebral hemispheres, additional details about location need to be included. 20 ul is a large volume to be injected in a non-ventricular location. How did the authors prevent bolus related tissue damage?

Response 5: We thank the reviewer for raising these points about Materials and Methods. We added the information in the lines 114, 116-117.

 Regarding the inoculation, we used free-hand inoculation, which is commonly used for intracerebral infection in several studies performed in the prion field and it is the elective method for bank vole and mice inoculation in our lab. Inoculation is performed frontally, in the frontal motor cortex and hippocampus. In the past, we observed using dyes (i.e. methylene blue) that the staining spreads in the tissue around the inoculation point and more rarely spreads on the surface of the cortex. Regarding the bolus-related tissue damage, we rarely (<1%) observed damage in inoculated brains.

Regarding the inoculated amount, Morton et al. 2001, suggest that injection volumes should not exceed 2% of brain volume.

Round 2

Reviewer 1 Report

The current version does not have new data to fortify the novelty of this work. The authors did modify their conclusion and supplied additional discussion. Scientifically, this work remains interesting but much to be done. This reviewer dose see the potential value of this kind of work, and the grave need of new animal models for Alzheimer's disease. Some of the "in progress" experiments mentioned by the authors' reply were very intriguing, and shall add significant novelty to this line of work.

Author Response

We thank the reviewer for his/her comments. Yes, we believe that we need new animal models for Alzheimer's disease. The development of a new wild-type model it's long and expensive work (in terms of time, scientific competence, and money) and includes multiple variables. The paper submitted includes the preliminary data regarding only tau from a single case of familial Alzheimer's disease, and we are conscious that accreditation of a new model in the complicated field of AD needs progressive steps. We hope to have more as fast as possible information to understand the value of bank vole as a new model for AD, and in particular to demonstrate the prion-like properties of A-beta.